# Characterizing Relevant MicroRNA Editing Sites in Parkinson’s Disease

**DOI:** 10.3390/cells12010075

**Published:** 2022-12-24

**Authors:** Chenyu Lu, Shuchao Ren, Wenping Xie, Zhigang Zhao, Xingwang Wu, Shiyong Guo, Angbaji Suo, Nan Zhou, Jun Yang, Shuai Wu, Yun Zheng

**Affiliations:** 1College of Landscape and Horticulture, Yunnan Agricultural University, Kunming 650201, China; 2Faculty of Life Science and Technology, Kunming University of Science and Technology, Kunming 650500, China; 3School of Criminal Investigation, Yunnan Police College, Kunming 650223, China

**Keywords:** RNA editing, Parkinson’s disease, hsa-miR-497-5p, OPA1, VAPB

## Abstract

MicroRNAs (miRNAs) are extensively edited in human brains. However, the functional relevance of the miRNA editome is largely unknown in Parkinson’s disease (PD). By analyzing small RNA sequencing profiles of brain tissues of 43 PD patients and 88 normal controls, we found that the editing levels of five A-to-I and two C-to-U editing sites are significantly correlated with the ages of normal controls, which is disrupted in PD patients. We totally identified 362 miRNA editing sites with significantly different editing levels in prefrontal cortices of PD patients (PD-PC) compared to results of normal controls. We experimentally validated that A-to-I edited miR-497-5p, with significantly higher expression levels in PD-PC compared to normal controls, directly represses OPA1 and VAPB. Furthermore, overexpression of A-to-I edited miR-497-5p downregulates OPA1 and VAPB in two cell lines, and inhibits proliferation of glioma cells. These results suggest that the hyperediting of miR-497-5p in PD contributes to enhanced progressive neurodegeneration of PD patients. Our results provide new insights into the mechanistic understanding, novel diagnostics, and therapeutic clues of PD.

## 1. Introduction

MicroRNAs (miRNAs) are small non-coding RNAs with about 22 nucleotides that normally repress their target mRNAs at the post-transcriptional level [1]. miRNAs have been shown to be important in Parkinson’s disease (PD). Kim et al. [2] found that miR-133b was specifically expressed in midbrain dopaminergic neurons and was deficient in midbrain tissue from patients with PD. miR-7 and miR-153 repressed α-synuclein (SNCA) expression in PD mouse models [3,4]. E2F1 and DP were repressed by let-7 and miR-184* [5]. Pathogenic LRRK2 inhibited let-7 and miR-184*, and consequently E2F1 and DP were upregulated [5]. miR-34b and miR-34c were downregulated in the amygdala, substantia nigra, frontal cortex, and cerebellum regions of PD patients, even in pre-motor stages [6]. A later study found that miR-34b and miR-34c could directly repress SNCA [7]. miR-205 suppressed expression of the LRRK2 protein and was significantly downregulated in the brains of patients with sporadic PD [8]. miR-30e was downregulated in the substantia nigra pars compacta of an MPTP-induced PD mouse model, and miR-30e directly repressed Nlrp3, which contributed to neuroinflammation [9]. miR-4639-5p was significantly upregulated in the plasma of PD patients and directly repressed DJ-1 (PARK7) [10]. Upregulation of miR-4639-5p resulted in oxidative stress and neuronal death [10]. miR-494 repressed DJ-1, which promoted neorodegeneration [11]. miR-494-3p was upregulated and repressed SIRT3 in an MPTP-induced PD mouse model, which caused motor impairment [12]. miR-124 directly repressed MEKK3, which potentially inhibited neuroinflammation in the development of PD [13]. miR-124-3p directly targeted ANAX5, which suggested a neuroprotective role of miR-124-3p in PD [14]. More evidences were reviewed in [15,16,17].

Some miRNAs are edited, such as Adenosine-to-Inosine (A-to-I) editing [18,19,20,21,22,23] performed by adenosine deaminase (ADAR) enzymes, and C-to-U editing performed by apolipoprotein B mRNA editing catalytic polypeptide-like (APOBEC) enzymes [24] during their biogenesis processes. Altered editing of miRNAs leads to human diseases, such as cancer [25,26,27,28]. Although A-to-I editing is prevalent in the brain [29,30,31,32] and the editing level gradually increases in the developmental procedure [21,32], the functional relevance of miRNA editing in PD is largely unknown.

To comprehensively characterize miRNA editing sites in brain tissues of PD patients, we analyzed 43 and 88 small RNA sequencing profiles of PD and control brain tissues, respectively. We identified 362 miRNA editing sites that had significantly different editing levels in the prefrontal cortices of PD patients (PD-PC) compared with the control group. The editing levels of five A-to-I and two C-to-U editing sites were significantly correlated with the ages of normal controls, which were disrupted in PD patients. One A-to-I editing site in miR-497-5p had significantly higher editing levels compared to normal controls in PD-PC samples and edited miR-497-5p directly repressed OPA1 and VAPB. We also demonstrated that overexpression of A-to-I edited miR-497-5p downregulated the expression of OPA1 and VAPB, and repressed proliferation of glioma cells. These results demonstrate that miRNA editing is severely disturbed and relevant in PD and offer novel insights into the etiology of PD. The abnormal miRNA editing patterns in PD might be used to develop novel diagnostic biomarkers and/or therapeutic targets for PD.

## 2. Material and Methods

### 2.1. The Small RNA Sequencing Profiles Used

To comprehensively identify miRNA mutation and editing (M/E) sites in PD, we collected 131 sRNA-seq profiles of postmortem PD patients and normal people from the NCBI SRA Database (Appendix A). These profiles included 29 prefrontal cortex samples of PD patients (PD-PC), 14 amygdalae samples of PD patients (PD-Am), 36 prefrontal cortex samples of normal controls (PC), 14 amygdalae samples of normal controls (Am), 6 frontal cortex samples of normal controls (FC), 6 corpus callosum samples of normal controls (CC), 2 inferior parietal lobe samples of normal controls (IPL), 2 temporal neocortex gray matter samples of normal controls (NG), 3 astrocyte cell lines of normal controls (As), and 19 unknown brain regions of normal controls (Unknown). The PD-PC and PC samples were from Brodmann Area 9 (BA9).

### 2.2. The Gene and Protein Expression Profiles Used

To understand the potential function of edited miRNAs, we examined the deregulated genes from brain samples of PD patients and normal controls. As listed in Appendix A, we selected 107 and 122 gene expression profiles of PD patients and normal controls, respectively, from seven cohorts, including two cohorts for the prefrontal cortex (Brodmann Area 9, briefly BA9) (GSE20168 and GSE68719), two cohorts for the substantia nigra (GSE7621 and GSE20292), one cohort for the putamen (GSE20291), one cohort for the cingulate gyrus (GSE110716), and one cohort for the subventricular zone (GSE130752).

A previous study reported proteomics profiles of prefrontal cortex (BA9) samples in 12 PD patients and 12 healthy people [33]. After comparing the proteomics profiles of the 12 PD patients to those of the 12 healthy people, 283 deregulated proteins were detected (limma package, corrected p<0.05) and used in our analysis.

### 2.3. Genome and Annotation of miRNAs Used

The unmasked human genomic sequence (hg38, GRCh38) was downloaded from the UCSC Genome Browser [34]. The bowtie-build program in the Bowtie package [35] was used to generate index files of the human genome. The pre-miRNA sequences and genomic positions of miRNAs in gff3 format were downloaded from miRBase (release 21) [36].

### 2.4. Analysis of Small RNA Sequencing Profiles

The selected sRNA-seq profiles were analyzed using the MiRME pipeline [37] with the default settings. Briefly, the raw reads whose sequencing scores of the first 25 nucleotides from the 5${}^{\prime }$ end had sequencing scores of 30 or higher were kept as qualified reads. Then, the unique sequences of the remaining reads were obtained and counts of unique reads with more than 18 nucleotides were calculated. Next, the unique reads were aligned to human pre-miRNAs using NCBI BLASTN [38] with the options of “-S 1 -m 8 -e 0.01” and the reads mapped to human pre-miRNAs were retrieved. Next, these reads mapped to pre-miRNAs were aligned to the genome using Bowtie (v1.0.0) [35] with the options of “-a -best -S -v 1”. Then, the alignments of reads to the genome were examined using the cross-mapping correction method [39] to adjust the weights or percentages of a unique read at each of its genomic loci. In the main step, the MiRME algorithm with the default parameters was used to identify M/E sites in miRNAs from the sequences and structures of pre-miRNAs, the alignment of reads to the genome generated by Bowtie, the reads mapped to pre-miRNAs, the alignments of reads to pre-miRNAs generated by BLASTN, and the results of the cross-mapping correction method [39].

The following criteria were used to define M/E sites with significant editing levels: (i) the relative level of editing was at least 5%; (ii) at least 10 reads supported the editing event; (iii) the score threshold of sequencing reads was 30; (iv) a multiple-test corrected *P*-value (using the Benjamini and Hochberg method [40]) smaller than 0.05; and (v) only M/E sites that had significant editing levels in 10% of all samples were kept for further analysis. Then, the obtained results of different samples were combined by a separate program in the MiRME package (see details in [37]). Based on the positions of M/E sites in miRNAs and mutations in dbSNP, the identified M/E sites were classified into nine different editing types, i.e., A-to-I, C-to-U, 3′-A, 3′-U, 3′-Other, 5′-editing, Other, SNP, and Pseudo [37].

To remove M/E sites due to random sequencing errors, 860 M/E sites that had significant editing levels in at least 10% of the selected samples (13 samples) were kept for further analysis. The 860 identified M/E sites were compared to reported editing sites in miRNAs in the DARNED database [41], the RADAR database [42], and in the literature [19,20,37,43,44,45]. Finally, the predicted M/E sites that belonged to A-to-I, C-to-U, and Other were manually examined.

All identified M/E sites were named by the names of the pre-miRNAs, positions of the M/E sites in the pre-miRNAs, the nucleotides from the reference pre-miRNA sequences in upper cases, and the edited/mutated nucleotides at the sites in lower cases. Additionally, edited miRNA was named by the pre-miRNA name, the position of the M/E site in pre-miRNA, and the edited/mutated nucleotide in lower case.

### 2.5. Comparing the M/E Sites to Reported SNPs

The identified M/E sites were compared to known SNPs in miRNAs organized in [46] (which was based on the dbSNP v137) and reported SNPs in dbSNP (v151). Only sites that satisfied the following criteria were regarded as SNPs: (i) they had the same genomic positions as the SNPs, (ii) had the same nucleotides as the alleles of the SNPs for both the original and changed nucleotides, and (iii) had editing levels of 100% in at least one of the 131 samples selected.

### 2.6. Identifying Conserved Editing Sites in miRNAs

The A-to-I and C-to-U editing sites were compared to their counterparts in *Macaca mulatta* [47] and *Mus musculus* [18,19]. The editing sites of the same editing types that were located on the same positions of mature miRNAs of at least two different species were considered as conserved editing sites.

### 2.7. Identifying Age-Related miRNA Editing Sites

The Pearson correlation, as well as its *p*-value, between editing level (in %) of each of the 860 editing sites in Appendix A and the age of death (y) was calculated with the corr function in MatLab (Mathworks, MA, USA) for the 29 PD-PC and 36 PC samples, respectively. The obtained *p*-values were corrected using the Benjamini and Horchberg method [40] using the mafdr function in MatLab (Mathworks, MA, USA). The editing sites with corrected *p*-values smaller than 0.05 were regarded as age-related.

### 2.8. Identifying M/E Sites with Significantly Different Editing Levels in PD

The editing levels of 860 M/E sites in the PD-PC and PC, and PC-Am and AM samples were compared, respectively, using the Mann–Whitney *U*-tests. The obtained *p*-values were corrected with the Benjamini and Hochberg method [40]. M/E sites with multiple test corrected *p*-values smaller than 0.05 were regarded as having significantly different editing levels in the PD-PC or PC-Am samples.

### 2.9. Clustering and Principle Component Analysis Using the Editing Levels of Selected M/E Sites

The editing levels of the 276 editing sites that had significantly different editing levels in PD-PC samples when compared to PC samples were used to perform hierarchical clustering with the hclust function in R. The editing levels were multiplied times 100 and added plus one, were then log2-scaled and used to calculate the correlation coefficient between samples, and one minus correlation coefficient values were used as distances between samples. The Ward D2 method was used in clustering analysis. The editing levels times 100 plus one of these 276 editing sites were log2-scaled and used to perform principle component analysis with the prcomp function in R.

### 2.10. Identifying Targets for Original and Edited miRNAs

M/E sites that satisfied the following criteria were chosen to identify the targets of original and edited miRNAs: (i) they had significantly different editing levels in PD-PC samples compared to PC samples; and (ii) were located in the seed regions of mature miRNAs. The targets of original and edited miRNAs were predicted using the MiCPAR algorithm [48] with its default parameters.

As listed in Appendix A, seven PAR-CLIP sequencing profiles prepared from HEK293 cells stably expressing FLAG/HA-tagged AGO proteins (AGO1, AGO2, AGO3, and AGO4) [49] were downloaded from the NCBI SRA database using the series accession number SRP002487. Four PAR-CLIP sequencing profiles prepared from HEK293 cell lines stably expressing HIS/FLAG/HA-tagged AGO1 or AGO2 [50] were downloaded from the NCBI SRA database using the series accession number SRP018015. Raw reads in these profiles were filtered to make sure that the first 25 nucleotides of the qualified reads had sequencing scores of 30 or higher. The 3${}^{\prime }$ adapters were cut for qualified reads. The remaining reads in these 11 profiles were combined and used in the identification of miRNA targets with the MiCPAR algorithm [48]. The annotation of NCBI RefSeq genes in the GTF file, the mRNA sequences of NCBI RefSeq genes (refMrna.fa.gz, version hg38), and soft-masked human genome sequences (version GRCh38) were downloaded from the UCSC Genome Browser [51] and used as inputs of the MiCPAR algorithm. The targets with at least one PAR-CLIP read with T-to-C variation were kept for further analysis.

### 2.11. GO and Pathway Analysis for the Original and Edited miRNAs

The GO (Gene Ontology) term and KEGG (Kyoto Encyclopedia of Genes and Genomes) pathway enrichment of the genes that were only targeted by the original or edited miRNAs were analyzed with KOBAS2, respectively [52]. The significantly enriched GO terms (with a multiple test corrected *p*-values smaller than 0.05) were divided into three major categories, i.e., Biological Process, Cellular Component, and Molecular Function. Then, the enriched GO terms and KEGG pathways of the original and edited miRNAs were compared.

### 2.12. Identifying Meaningful Targets for Edited miRNAs

A previous study [33] identified 1095 deregulated genes using the Wald tests (in DESeq2 [53]) by analyzing RNA-seq data sets for 19 PD and 24 control samples of the prefrontal cortex (Brodmann Area 9, BA9). This study also examined the protein abundance levels of BA9 samples of 12 PD patients and 12 normal people using MS3 proteomics and identified 283 proteins with significantly different levels (limma package, corrected p<0.05) in PD [33]. Another study [54] generated 30 gene expression profiles of BA9 with microarray, with 15 profiles for both PD and PD-PC. The deregulated genes were identified with the limma package.

Because these RNA-seq and proteomics profiles were also from the BA9 regions, which were the same as the regions of PD-PC and PC sRNA-seq profiles in this study, these deregulated genes (mRNAs) and proteins were used to identify meaningful targets for edited miRNAs. For edited miRNAs that had higher editing levels and normalized abundances in the PD-PC samples, we chose their targets, identified with the MiCPAR algorithm, which showed lower expression levels in PD-PC samples, and vise versa.

### 2.13. Validating Selected Targets of Edited hsa-miR-497-5p

The 3${}^{\prime }$ UTR segments of human OPA1/VAPB, mutated OPA1/VAPB, crab-eating monkey (*Macaca fascicularis*) OPA1/VAPB, mutated monkey OPA1/VAPB, and mouse (*Mus musculus*) Opa1/Vapb containing the hsa-miR-497_25g complementary sites (∼130 nt with 70 nt on both sides) were synthesized and cloned into the plasmid pmirGLO (Promega, Madison, WI, USA) with XholI and SalI sites, and named as pGLO-OPA1/VAPB, pGLO-OPA1m/VAPBm, pGLO-mfaOPA1/mfaVAPB, pGLO-mfaOPA1m/mfaVAPBm, and pGLO-musOpa1/musVapb, respectively. The genomic regions of hsa-pre-mir-497 and hsa-pre-mir-497_25g were synthesized and cloned into the plasmid pCDNA 3.1(+) (Invirtrogen, Carlsbad, CA, USA) with HindIII and BamHI sites, and named as p497 and p497_25g, respectively. All of the plasmids were validated by Sanger sequencing (ABI3730, Thermo Fisher, Waltham, MA, USA).

A human 293T cell line was cultured in DMEM high glucose medium with 10% FBS and 1% NEAA. A total of 500 μL medium was inoculated to each well of a 24-well plate. Totals of 2 μg p497_25g and 2 μg pGLO-OPA1 were added to one well as a group of co-transfection. Similarly, one pGLO plasmid and one pCDNA were co-transfected to 293T cells, respectively. Then, 2 μL of DNA-INVI DNA Transfection Reagent (Invigentech, Carlsbad, CA 92008, USA) was added and mixed into each well. In the experiments, each group had three biological replicates. The TransDetect double-luciferase Reporter Assay Kit (Beijing TransGen Biotech, Beijing, China) was used to detect luciferase activities for each of the biological replicates with three technical repeats. The mean value of the three technical repeats was used as the value of one biological replicate. The luciferase activities of different groups were then compared with two-tailed *t*-tests.

### 2.14. Conservation Analysis of Edited miR-497-5p Complementary Sites

The conservation scores (PhyloP scores [55] for 30 mammals) of the A-to-I edited miR-497-5p complementary sites on OPA1 and VAPB were downloaded from the UCSC Genome Browser [34]. The sequences of the A-to-I edited miR-497-5p complementary sites on OPA1 and VAPB in 30 mammals were downloaded from the UCSC Genome Browser and used to build phylogenetic trees with ClustalX (v2.1) [56]. The obtained phylogenetic trees were visualized with TreeView (v1.6.6) [57].

### 2.15. Cell Culture

Human glioma cell lines (U-118-MG and Hs683) were purchased from the Cell Bank of the Kunming Institute of Zoology, Chinese Academy of Sciences (Kunming, China). All cells were cultured in Dulbecco modified Eagle’s medium (DMEM) (Corning), supplied with 10% FBS and 1% penicillin/streptomycin solution. The cell culture was maintained in a 37 ${}^{\circ }$C incubator with 5% CO${}_{2}$.

### 2.16. Cell Proliferation Assay

To assess the cell proliferation activities, cell groups were gathered, resuspended in 10% FBS medium ($2\times 10^{4}$ cells/mL), seeded in a 96-well plate and then incubated in a 37 ${}^{\circ }$C incubator with 5% CO${}_{2}$. Then, cells were transiently transfected with the p497 and p497_25g plasmids, using Lipofectamine 2000 (Invitrogen, Carlsbad, CA, USA) according to the manufacturer’s protocol. Then, after 24, 48, and 72 h of culture, each well received 10 μL of the Cell Counting Kit-8 solution (Beijing TransGen Biotech, Beijing, China), and was subsequently incubated at 37 ${}^{\circ }$C for 1 h. The proliferation activities of cells were measured using an absorbance microplate reader (PHERAstar FS, BMG LABTECH), at wavelengths of 490 and 450 nm. The cell proliferation activities of different groups were then compared with two-tailed *t*-tests.

### 2.17. Rna Isolation and Quantitative Real-Time Polymerase Chain Reaction

Total RNAs were extracted from the cells 48 h after p497_25g plasmid treatment using the TriQuick total RNA extraction Reagent Kit (Solarbio, Beijing, China). The first strand complementary DNAs (cDNAs) were generated with the Servicebio RT First Strand cDNA Synthesis Kit (Servicebio, Wuhan, China) for OPA1 and VAPB, according to the manufacture’s protocols. Real-time PCR evaluation of mRNAs was performed using the 2xSYBR Green qPCR Master mix PCR Kit (Servicebio, Wuhan, China). The qRT-PCR was performed on a Step one plus Real-time PCR System (Applied Biosystems, ABI). The PCR primer sequences are provided in Appendix A. The relative RNA expression was examined using the $2^{-\Delta \Delta Ct}$ method [58]. The expression levels of OPA1 and VAPB in different groups were then compared using two-tailed *t*-tests.

### 2.18. Role of the Funding Source

The funders had no role in study design, data collection and analysis, decision to publish, or preparation of the manuscript.

## 3. Results

### 3.1. An Overview of Identified Editing Sites in miRNAs

We used the MiRME pipeline [37] to analyze the 131 sRNA-seq profiles selected with the default settings. Totally, we found 860 significant M/E sites (as shown in Figure 1 and Appendix A) by employing the criteria of at least 10 reads supporting the M/E sites, multiple test-corrected *p*-values smaller than 0.05, and the M/E sites being identified in at least 10% of the selected sRNA-seq profiles. The largest category of these M/E sites was 3${}^{\prime }$-A, accounting for 43.8% (377 sites), followed by 3${}^{\prime }$-U (41.7%), 3${}^{\prime }$-Other (5.5%), and 5${}^{\prime }$-editing (2.7%) (Figure 1a and Appendix A). A-to-I and C-to-U editing sites accounted for 2.7% (23 sites) and 0.5% (4 sites), respectively (Figure 1a). We found 14 SNPs after comparing the identified M/E sites to SNPs in dbSNP (Figure 1a). The Other type of editing sites accounted for 0.5% (four sites) (Figure 1a).

The A-to-I, C-to-U, and Other sites were further classified based on their types of variations (Appendix A). Similar to previous work [37], there could be several 3${}^{\prime }$-editing sites in one pre-miRNA, but most pre-miRNAs only had 1 or 2 central sites and 5${}^{\prime }$ sites in our selected samples (Appendix A). However, some miRNAs may have had a few editing events at the 5’ end in our previous study [37].

### 3.2. Age-Related Editing Sites in PD and Normal Controls

Because previous studies reported that A-to-I editing level was increasing during the developmental process [21,32], we examined the Spearman correlation (ρ) between editing levels of the 860 editing sites and the ages (at death) of 36 PC and 29 PD-PC samples, respectively. There were 227 sites with significant ρ in PC, but only 60 sites with significant ρ in PD-PC samples (Figure 1b and Appendix A). When these sites were separated as ones with positive and negative ρ values, the numbers of sites with significant ρ values in PD-PC samples were also much smaller than those in PC samples (Figure 1b and Appendix A). The numbers of types of editing sites were decreased in PD-PC samples too. For example, there were no A-to-I sites with positive ρ values in the PD-PC samples (Appendix A). Furthermore, the Kullback–Leibler divergences of the distributions of different types of editing sites in PC and PD-PC samples were 1.81, 2.48, and 3.5 for all, sites with positive ρ values, and sites with negative ρ values, respectively (Appendix A). These results suggest that the gradually increasing or decreasing editing levels of miRNAs during the aging procedure were severely disturbed in PD.

The ρ values of 23 A-to-I and 4 C-to-U editing sites are carefully examined in Figure 1c and Appendix A. In the PC samples, there were three and one A-to-I sites with significant positive and negative ρ values, respectively (Figure 1c–e). In comparison, only one A-to-I site had a significant negative ρ value in PD-PC (Figure 1c,d). The median value of ρ of A-to-I sites in PC was slightly larger than 0 (Figure 1c), which is consistent with the ever-increasing A-to-I editing level in the developmental process noticed previously [21,32]. However, the median ρ value of A-to-I sites in PD-PC samples was smaller than 0 (Figure 1c), further suggesting the disrupted A-to-I editing of miRNAs in PD. By contrast, the C-to-U editing sites had a weak negative median ρ value in PC (Figure 1c), suggesting that C-to-U editing sites of miRNAs generally had decreasing editing levels when people were aging. Two C-to-U editing sites (see Figure 1c,d,f) had significant ρ values in PC.

### 3.3. A-To-I Editing Sites

We found 23 significant A-to-I editing sites in total from the selected samples (Appendix A). At least 14 of these 23 sites were conserved in other mammals [19,47] (Appendix A). Similar to previous results [19,20,37,47], the 5${}^{\prime }$ and 3${}^{\prime }$ sides of these A-to-I editing sites preferred to be U and G, respectively (Appendix A). For example, two identified A-to-I editing sites are shown in Appendix A.

### 3.4. C-To-U Editing Sites

We carefully explored the four C-to-U editing sites (Appendix A). Three of these four C-to-U editing sites were conserved in primates after being compared with results of monkey [47] (see Appendix A). The neighboring nucleotides of these C-to-U sites preferred to be C on both the 5${}^{\prime }$ and 3${}^{\prime }$ sides (Appendix A), consistent with the CCC motif of APOBEC3G [59]. Three examples of C-to-U editing sites are shown in Appendix A.

### 3.5. Identified SNPs in miRNAs

By comparing the M/E sites to SNPs reported in dbSNP and examining their editing levels, 14 M/E sites were regarded as SNPs (as shown in Appendix A). Three of the fourteen SNP sites are shown in Appendix A. The editing levels of these three sites were 100% in the selected samples (Appendix A).

### 3.6. Relevant miRNA Editing Sites in PD

To identify editing sites that were relevant in PD, we compared the editing levels of 860 significant editing sites in PD-PC and PC, and PD-Am and Am samples. We obtained 276 M/E sites that had significantly different editing levels in PD-PC samples compared to PC samples (Appendix A). Most of these 276 sites (63.04% or 174 sites) had increased editing levels in PD-PC samples, and 36.96% of these sites had decreased editing levels in PD-PC samples (Appendix A). Three A-to-I editing sites (hsa-mir-497_25_A_g, hsa-mir-1251_10_A_g, and hsa-mir-1301_52_A_g) had increased editing levels in PD-PC samples, and two A-to-I editing sites (hsa-let-7a-2_28_A_g and hsa-mir-411_20_A_g) showed decreased editing levels in PD-PC samples compared to PC samples (Figure 2a and Appendix A). Three C-to-U and one Other editing sites had decreased and increased editing levels in PD-PC samples compared to PC samples, respectively (Figure 2b). There were no M/E sites with significantly different editing levels in the comparison between PD-Am and Am samples.

We used the editing levels of these 276 M/E sites to perform principle component analysis and hierarchical clustering for the PD-PC and PC samples. The results show that although PD-PC samples were grouped together, 12 PC samples were clustered together with the PD-PC samples (Figure 2c,d). As indicated by the arrow in Figure 2c, some PC samples seem to be gradually falling to the cluster of PD-PC samples. These results suggest that although these 12 PC samples were clinically normal, their miRNAs showed similar editing patterns to those of PD patients. Therefore, we treated these 12 PC samples as PD-PC samples (named as PD-PC-like samples). Seven of the nine editing sites shown in Appendix A demonstrated significantly different editing levels (corrected p<0.05, Mann–Whitney *U*-tests) when the 12 PD-PC-like samples were compared to other PC samples, suggesting that these 12 PD-PC-like samples had different miRNA editing patterns compared to other PC samples (see Appendix A).

To check the age difference of PD-PC-like samples, we carefully examined the editing levels of hsa-mir-497_25_A_g (see Figure 2e) and ages of PC, PD-PC-like, and PD-PC samples. As in Figure 2e, PD-PC-like samples had higher editing levels for hsa-mir-497_25_A_g (corrected $p=1.9\times 10^{-4}$, Mann–Whitney *U*-test, see Appendix A) and larger ages than other PC samples ($p=1.1\times 10^{-3}$, Student’s *t*-test, Figure 2f). In comparison, the ages of PD-PC-like and PD-PC samples had no significant difference (p=0.69, Student’s *t*-test, Figure 2f).

### 3.7. Target Analysis of A-to-I Editing Sites in Seeds

We selected four A-to-I sites that located in the seed regions of mature miRNAs (Appendix A) and identified the targets of A-to-I edited and original miRNAs with the MiCPAR pipeline [48], as listed in Appendix A. In the MiCPAR analysis, 11 PAR-CLIP sequencing profiles were used and only targets with at least 1 PAR-CLIP read with T-to-C variation were kept for further analysis (see Materials and Methods for details). After comparing the targets of A-to-I edited and original miRNAs, we found that these A-to-I editing events severely changed the target sets of these miRNAs (Appendix A), as well as the enriched GO terms and KEGG pathways of these targets (Appendix A).

Since A-to-I edited hsa-miR-497-5p (hsa-mir-497_25g) had a much higher expression level than the other three A-to-I edited miRNAs (Appendix A), we next carefully examined the targets of hsa-mir-497_25g. After comparing the predicted targets of edited miRNAs with the deregulated genes and proteins in prefrontal cortex (BA9) samples (Figure 3a and Appendix A, see details in Materials and Methods), we found that hsa-mir-497_25g targeted OPA1 mitochondrial dynamin-like GTPase (OPA1) and VAMP-associated protein B and C (VAPB) (Figure 3b). Presumably due to the upregulation of hsa-mir-497_25g (Figure 3c) in PD-PC samples, OPA1 and VAPB were significantly downregulated in PD-PC samples (Figure 3d,e). The protein expression levels of OPA1 and VAPB were also significantly downregulated in PD-PC samples (corrected p<0.05, as in [33]).

Furthermore, OPA1 was significantly downregulated in the substantia nigra (SN) (Appendix A), and was mildly downregulated in the putamen of PD patients (Appendix A). However, in the subventricular zone (SZ) and cingulate gyrus, the expression of OPA1 did not change significantly (Appendix A). In SN, VAPB had a weak decreasing trend in PD patients (Appendix A, but had no change in another cohort (Appendix A) and in the putamen, SZ and cingulate gyrus (Appendix A, respectively).

Importantly, consistent with the significant positive correlation between the editing level of hsa-mir-497_25_A_g and ages of individuals in PC (Figure 1d), the expression levels of OPA1 and VAPB were significantly negatively correlated with the ages of individuals in PC (Figure 3f,g and Appendix A) and potentially in SN as well (ρ=−0.36 and −0.47, respectively, see Appendix A, respectively). The Spearman correlation ρs between the expressions of OPA1 and VAPB and ages were weak and insignificant in putamen (Appendix A, respectively), and were positive in SZ (ρ=0.29 and 0.60 in Appendix A, respectively). In PD, these correlations were generally weakened (Appendix A for OPA1 and VAPB, respectively), except in putamen, where the negative correlations were slightly enhanced (Appendix A for OPA1 and VAPB, respectively).

### 3.8. hsa-mir-497_25g Directly Represses OPA1 and VAPB

The complementary sites of hsa-mir-497_25g on OPA1 and VAPB were only partially conserved in mammals (Figure 4a–d and Appendix A). For both OPA1 and VAPB, in the 8mer opposite to the seed of edited miR-497-5p (red rectangles in Figure 4a,b), a few nucleotides had negative PhyloP conservation scores [55], indicating that these two sites were fast-evolving in primates. Even within primates, four and seven species did not have conserved sequences complementary to the seed of hsa-mir-497_25g for OPA1 and VAPB, respectively (Figure 4c,d, respectively).

To verify that hsa-mir-497_25g directly inhibited OPA1 and VAPB, we performed luciferase assays in 293T cells for hsa-mir-497_25g, original hsa-mir-497, OPA1/VAPB, mutated OPA1/VAPB (see Figure 4a,b, respectively). As in Figure 4e,f, hsa-mir-497_25g induced significant decreases of luciferase activities for OPA1 and VAPB, indicating that hsa-mir-497_25g directly repressed OPA1 and VAPB, respectively. In comparison, when being co-transfected with mutated OPA1 and mutated VAPB, hsa-mir-497_25g could not repress mutated OPA1 and mutated VAPB (Figure 4e,f, respectively). The original miR-497-5p could not repress OPA1 and VAPB too (Figure 4e,f, respectively).

The complementary sites of hsa-mir-497_25g on OPA1 and VAPB were only conserved in some primates including monkey, but non-conserved in rodents (Figure 4a–d). As expected, A-to-I edited miR-497-5p could repress OPA1 and VAPB of monkey (Appendix A), but could not repress Opa1 and Vapb of mouse (Figure 4g–h).

To summarize, hsa-mir-497_25g directly suppressed OPA1 and VAPB in human and in some non-human primates, but not in mouse and those primates with non-conserved complementary sites of hsa-mir-497_25g. The non-conserved complementary sites of hsa-mir-497_25g might contribute to the advanced functions and fast evolving of prefrontal cortex of human.

### 3.9. hsa-mir-497_25g Suppresses the Proliferation of Glioma Cells

To further validate that hsa-mir-497_25g repressed OPA1 and VAPB, we transfected p497_25g into two glioma cell lines, i.e., U-118-MG and Hs683, and examined the expression levels of OPA1 and VAPB with quantitative RT-PCR (qRT-PCR) experiments. The results of qRT-PCR experiments showed that transfection of p497_25g plasmid downregulated the expression levels of OPA1 and VAPB in the two cell lines, respectively (Figure 5a,d and Appendix A).

To explore the role of hsa-mir-497_25g in glioma cell line growth, we transfected U-118-MG and Hs683 with p497 and p497_25g plasmids, respectively. Then, the cell lines were subjected to CCK-8 assay of proliferation. We measured the proliferation activities of the cells after transfection of the plasmids at 24, 48, and 72 h. U-118-MG cells without transfection of plasmids and treated with Lipofectamine 2000 were also measured as controls. The results showed that miR-497_25g had a noticeable inhibitory effect on the proliferation of U-118-MG cells when compared to untreated cells, cells treated with Lipofectamine 2000, and cells transfected with p497 (Figure 5e–g). As shown in Figure 5h,i, miR-497_25g also had a noticeable inhibitory effect on the proliferation of Hs683 cells when compared to untreated cells and cells transfected with p497. Snapshots of the cells as controls and cells transfected with p497_25g are also shown in Figure 5k,l, respectively. In summary, these results indicate that transfection of the miR-497_25g plasmid significantly represses the proliferation of U-118-MG cells.

## 4. Discussion

By analyzing small RNA sequencing profiles of brain tissues of 43 PD patients and 88 normal controls, we identified 362 miRNA editing sites with significantly different editing levels in the prefrontal cortices of PD patients (PD-PC) compared to results of normal controls. Although a few studies reported A-to-I editing sites of miRNAs in brains of normal people, our results provide a comprehensive view of the miRNA editome with different types of editing events in brain tissues of PD for the first time, which significantly increases our knowledge about miRNA editing in PD.

The ever increasing editing levels of A-to-I editing during developmental process has been noticed in several previous studies [21,32]. Similarly, we found that the editing levels of three A-to-I editing sites in miRNAs were significantly positively correlated with ages in normal PC samples and these significant correlations were disturbed in PD-PC samples (Figure 1d). Furthermore, our results demonstrate that the editing levels of two C-to-U editing sites (see Figure 1c,f) were negatively correlated with the ages of normal controls, which also disappeared in PD patients. These results suggest potential functional relevances of these C-to-U editing sites in PD.

*OPA1* localizes to the inner mitochondrial membrane and plays roles in regulating mitochondrial stability and energy output. Mutations in OPA1 have been associated with optic atrophy type 1 [60,61]. Recently, the association between OPA1 and parkinsonism was also noticed [62,63]. One recent study found that three members of a family had autosomal dominant optic atrophy caused by OPA1 mutations and two of them developed nonsyndromic PD [64]. OPA1 was downregulated in PD-PC (BA9) (Figure 3d,e) and in the substantia nigra of PD patients (Appendix A). PD patients carrying the G2019S mutation showed decreased levels of mature OPA1 [65]. OPA1 haploinsufficiency led to progressive loss of iPSC-derived dopaminergic neurons [66]. Autosomal dominant LRRK2 mutations were associated with both familial and sporadic PD [65,67]. LRRK2 directly interacted with OPA1, presumably to regulate membrane dynamics important for mediating endocytosis and mitochondrial function [65].

Neuronal synaptic dysfunction is one of the key features of Parkinson’s disease [68,69]. VAPB protein is a membrane protein found in plasma and intracellular vesicle membranes, and is involved in vesicle trafficking. VAPB was downregulated in BA9 of PD-PC (Figure 3d,e). Downregulation of VAPB and its interacting protein PTPIP51 reduced dentritic spine numbers and synaptic activity [70]. A missense mutation (P56S) in VAPB was reported to be associated with motorneuron degeneration in affected amyotrophic lateral sclerosis (ALS) patients [71]. Transgenic mice that expressed human P56S *VAPB* developed progressive hyperactivities and other motor abnormalities, and showed progressive loss of corticospinal motor neurons [72]. By contrast, overexpression of VAPB in mouse ALS model slowed the motor impairment and promoted the survival of spinal motor neurons [73].

α-synuclein is a PD-related protein that localizes to the nerve terminal [74] and directly binds to VAPB [74] and VAMP2 [75]. Overexpression wild-type and familial Parkinson’s disease mutant α-synuclein disrupt the interaction between VAPB and PTPIP51 to loosen ER–mitochondria associations [74]. The downregulation of VAPB in PD patients may exaggerate the toxic effect of α-synuclein.

Furthermore, we showed that transfection of p497_25g plasmid significantly downregulated the expression levels of OPA1 and VAPB in two cell lines (Figure 5a–d). Additionally, tranfection of miR-497_25g had significant inhibitory effects on the proliferation of U-118-MG and Hs683 cells compared to controls (Figure 5e–j).

In summary, hsa-mir-497 was edited at a normal level and A-to-I edited miR-497-5p repressed OPA1 and VAPB to induce neorodegeneration at a normal pace in healthy PC samples (Figure 6a). In comparison, the editing level of hsa-mir-497_25_A_g was significantly increased in PD-PC samples and the A-to-I edited miR-497-5p dominantly repressed OPA1 and VAPB, which led to significant downregulation of OPA1 and VAPB in PD-PC (Figure 6b). Consequently, the significant repression of OPA1 and VAPB contributed to the enhanced neurodegeneration in PD. These results suggest that novel therapies of PD might be designed by either over-expressing OPA1/VAPB or repressing the editing level of hsa-mir-497_25_A_g. Furthermore, the significant correlation between ages of normal people and editing level of hsa-mir-497_25_A_g, as well as expression levels of OPA1 and VAPB, and the significantly increased editing level of hsa-mir-497_25_A_g in PD might be used to develop novel diagnostic methods of PD.

## Figures and Tables

**Figure 1 cells-12-00075-f001:**
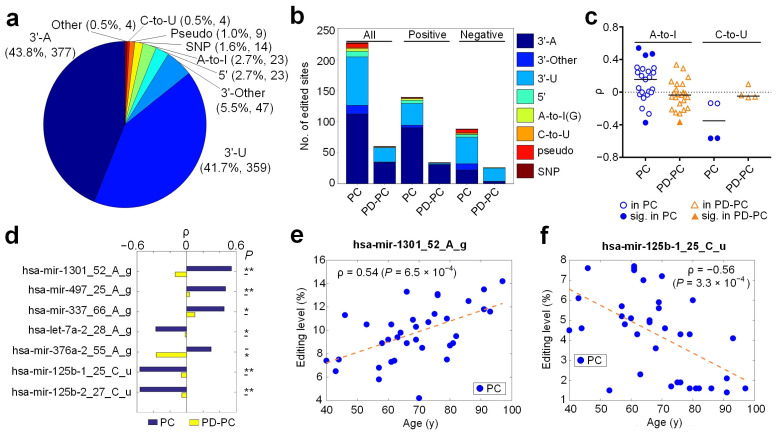
A summary of the identified miRNA mutation and editing sites. (**a**) The categories of significant M/E sites in miRNAs. (**b**) The numbers of different types of editing sites that had significant Spearman correlation (ρ) between editing levels and the ages of individuals in the PC and PD-PC groups. (**c**) The distributions of ρ for A-to-I and C-to-U sites in the PC and PD-PC samples, respectively. The solid markers are editing sites whose *p*-values of ρs are smaller than 0.05. The black lines represent the mean values of ρs of these editing sites. (**d**) Seven selected editing sites with significant ρ values in either PC or PD-PC samples. *: p<0.05; **: corrected p<0.01. (**e**) Plots of ages of PC samples against the editing level of hsa-mir-1301_52_A_g. (**f**) Plots of ages of PC samples against the editing level of hsa-mir-125-1_25_C_u. In Part (**e**,**f**), the dashed lines represent linear fits of the points. See also Appendix A for source data.

**Figure 2 cells-12-00075-f002:**
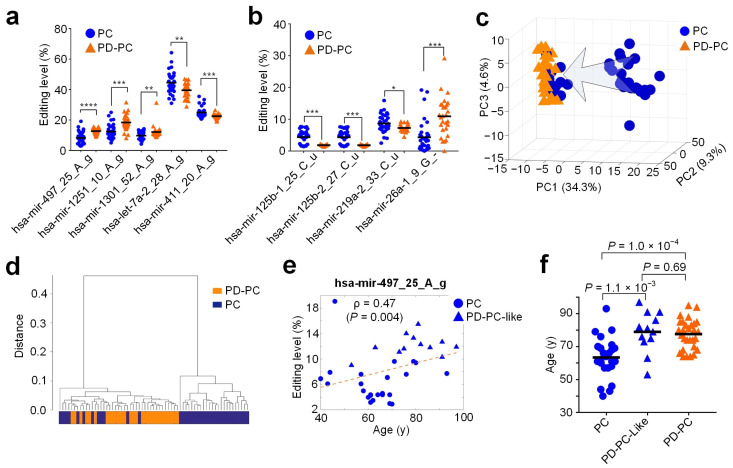
M/E sites that had significantly different editing levels in PD-PC samples when compared to PC samples. (**a**,**b**) Five A-to-I and four other types of M/E sites with significantly different editing levels in PD-PC. *: corrected p<0.05, **: corrected p<0.01, ***: corrected p<0.001, and ****: corrected p<0.0001, Mann–Whitney *U*-test. (**c**,**d**) PCA and clustering analysis using the editing levels of 276 M/E sites with significantly different editing levels in PD-PC. (**e**) The plot of ages against the editing levels of hsa-mir-497_25_A_g in PC samples and PD-PC-like samples. The dashed line represents linear fit of the points. (**f**) Comparisons of ages of PC, PD-PC-like, and PD-PC samples. *p*-values were based on two-tailed *t*-tests. The black lines represent the mean values. See also Appendix A for source data.

**Figure 3 cells-12-00075-f003:**
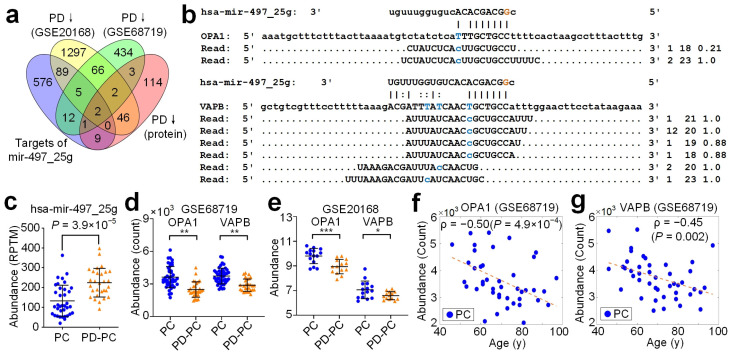
The target analysis of A-to-I edited hsa-miR-497-5p. (**a**) Integrated analysis of targets of edited hsa-miR-497-5p (hsa-miR-497_25g) and the downregulated genes and proteins in PD-PC (BA9). (**b**) The complementary sites of hsa-miR-497_25g on OPA1 and VAPB, and the PAR-CLIP sequencing reads from these sites. The three numbers after the reads were the raw frequencies, length (in nt), and the weights of reads at the locus. (**c**) Comparison of abundances of hsa-miR-497_25g in PC and PD-PC. (**d**,**e**) Comparison of abundances of OPA1 and VAPB in PC and PD-PC (BA9). *: corrected p<0.05; **: corrected p<0.01; ***: corrected p<0.001, DESeq2 and limma package. (**f**,**g**) The plots of the ages of PC samples against the abundances of OPA1 and VAPB. In (**f**,**g**), the dashed lines represent linear fits of the points. See also Appendix A for source data.

**Figure 4 cells-12-00075-f004:**
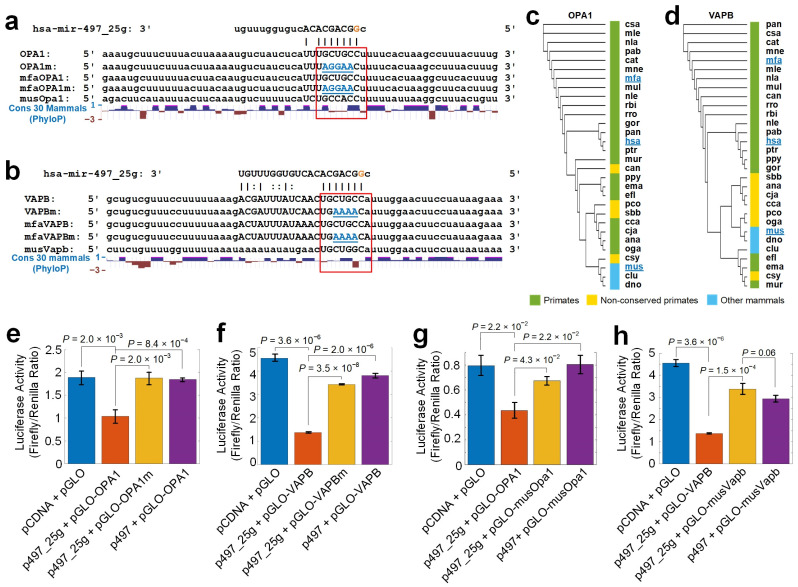
Validating selected targets of A-to-I edited miR-497-5p. (**a**,**b**) The segments of 3${}^{\prime }$ UTRs of OPA1 and VAPB (OPA1 and VAPB), mutated segments of 3${}^{\prime }$ UTRs of OPA1 and VAPB (OPA1m and VAPBm), segments of 3${}^{\prime }$ UTRs of monkey OPA1 and VAPB (mfaOPA1 and mfaVAPB), mutated segments of 3${}^{\prime }$ UTRs of monkey OPA1 and VAPB (mfaOPA1m and mfaVAPBm), and segments of 3’ UTRs of mouse Opa1 and Vapb (musOpa1 and musVapb). The conservation scores for 30 mammals in these regions were calculated with PhyloP. The regions in the red rectangles were 3${}^{\prime }$-UTR parts opposite to the seed region of hsa-mir-497_25g. The blue nucleotides in OPA1m, mfaOPA1m, VAPBm, and mfaVAPBm were the mutated nucleotides. (**c**,**d**) The phylogenetic trees of A-to-I edited miR-497-5p complementary sites on OPA1 and VAPB in 30 mammals. The abbreviations of the 30 species are listed in Appendix A. The three species with blue names, i.e., human (hsa, *Homo sapiens*), monkey (mfa, *Macaca fascicularis*), and mouse (mus, *Mus musculus*) were selected for luciferase experiments. (**e**–**h**) The luciferase activities when co-transfecting a pGLO plasmid of human and mouse 3${}^{\prime }$ UTRs in Part (**a**,**b**) and a pCDNA plasmid containing original pre-hsa-mir-497 (p497) or pre-hsa-mir-497_25g (p497_25g), respectively. The values shown were mean values ± SDs. *p*-values were based on two-tailed *t*-tests. See also Appendix A for monkey results and Appendix A for source data.

**Figure 5 cells-12-00075-f005:**
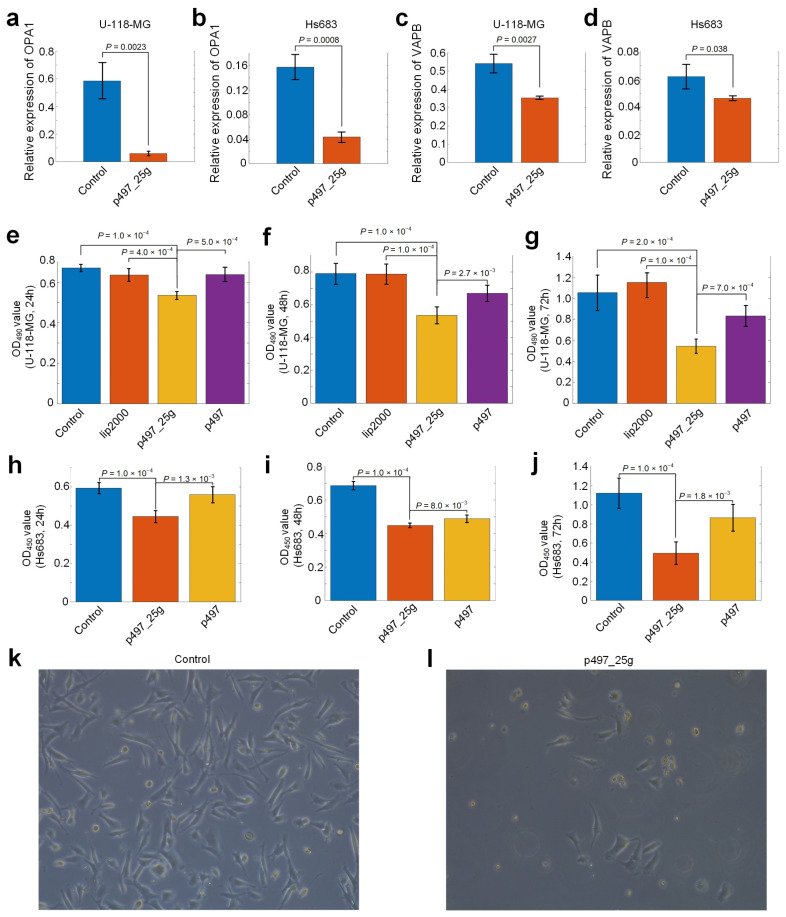
The expression levels of OPA1 and VAPB in glioma cell lines and proliferation of U-118-MG and Hs683 cells transfected with p497_25g. (**a**–**d**) The expression levels of OPA1 and VAPB in U-118-MG and Hs683 cells transfected with p497_25g plasmid compared with controls. (**e**–**g**) The cell proliferation of U-118-MG cells transfected with p497_25g, p497, treated with Lipofectamine 2000 only (lip2000), and control cells (without transfection and without treatment of Lipofectamine 2000). (**h**–**j**) The cell proliferation of Hs683 cells transfected with p497_25g, p497, and control cells (without transfection and without treatment of Lipofectamine 2000). The proliferation of cells was examined using the CCK-8 assay. (**k**) The morphology of cells in the control group after 72 h observed under an electron microscope with a diameter of 200 μm. (**l**) The morphology of cells in groups treated with the p497_25g plasmid after 72 h observed under an electron microscope with a diameter of 200 μm. In (**a**–**j**), *p*-values were based on two-tailed *t*-tests.

**Figure 6 cells-12-00075-f006:**
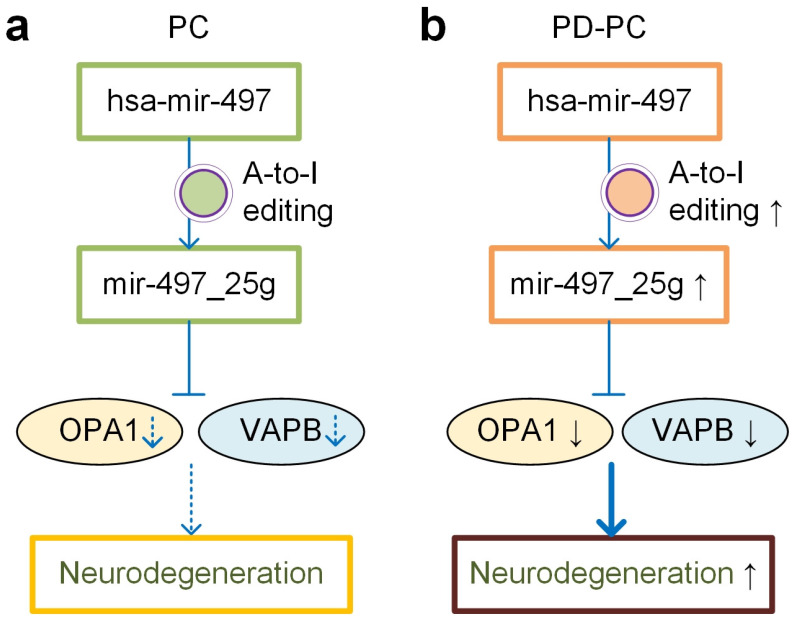
The role of hsa-mir-497_25_A_g in PD. (**a**) In PC tissue, the A-to-I editing of hsa-mir-497 is performed at a normal level. The miR-497_25g with a normal expression level represses OPA1 and VAPB to induce neurodegeneration at a normal pace. (**b**) The A-to-I editing of hsa-mir-497 increases in PD-PC tissue. The higher expression level of miR-497_25g leads to a significant downregulation of OPA1 and VAPB, which consequently contributes to enhanced neurodegeneration in PD-PC tissue.

## Data Availability

The 131 sRNA-seq profiles, 7 cohorts of gene expression profiles, and 11 PAR-CLIP sequencing profiles are available in the NCBI GEO and SRA databases under the accession numbers listed in Appendix A.

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
