# Peer review of "Characterizing Relevant MicroRNA Editing Sites in Parkinson’s Disease"

_cells, 2022, doi:10.3390/cells12010075_

Round 1
Reviewer 1 Report
The manuscript entitled "Characterizing relevant microRNA editing sites in Parkinson’s
disease" is well written and describes a rational approach to identify potential deregulated miRNA editing PD.
Two major concern:
OPA1 and VAPB where selected by overlapping different datasets. Since experiments were involved in the study it is unclear why the authors choose to confirm the predicted targets OPA1 and VAPB and not identify genome wide targets by RNA-Seq of 497 and p497_25g transfected Human glioma cell lines. They already have the experiment and the RNA. They can globally see the changes of predicted 497 and p497_25g target mRNAs, confirm their entire hypothesis and choose the most potent targets of p497_25g.
Author Response
Responses to Reviewers’ Comments
Reviewer 1 Comments
Comments 1.1 The manuscript entitled "Characterizing relevant microRNA editing sites in Parkinson’s disease" is well written and describes a rational approach to identify potential deregulated miRNA editing PD.
Response 1.1
Thank you.
Comments 1.2. OPA1 and VAPB where selected by overlapping different datasets. Since experiments were involved in the study it is unclear why the authors choose to confirm the predicted targets OPA1 and VAPB and not identify genome wide targets by RNA-Seq of 497 and p497_25g transfected Human glioma cell lines. They already have the experiment and the RNA. They can globally see the changes of predicted 497 and p497_25g target mRNAs, confirm their entire hypothesis and choose the most potent targets of p497_25g.
Response 1.2.
We did not choose to identify the targets of miR-497_25g by performing RNA-Seq profiles of p497 and p497_25g transfected Human glioma cell lines because the glioma cell lines were very different from the brain tissue samples of PD patients and normal people. The genes expressed in glioma cell lines were different from those expressed in neurons in brain tissue samples of PD patients and normal people. The OPA1 and VAPB were chosen in the experiments because these two genes were significantly downregulated in brain tissues of PD patients compared to normal controls, as repeatedly shown in different batches of data sets.
Reviewer 2 Comments
Comments 2.1 this manuscript contains a huge amount a data analysis and develops noval approaches to investigate the impact of miRNA editing on gene expression in the fram of Parkinson's disease.
Response 2.1
Thank you.
Comments 2.2 The only limitation that I see would be, in the introduction, to provide only the relevant examples of miRNAs likely involved in PD, rather that this exhaustive list with 13 references. Otherwise, the paper is very impressive.
Response 2.2
We included these references because the miRNAs reported in these papers were very relevant in PD. Although the exact reason of PD is still unclear, some key genes such as SNCA and LRRK2 are widely reported to be involved in PD. Therefore, we reviewed the miRNAs that were involved in the regulatory networks of these key genes of PD in the Introduction of our manuscript.
Comments 2.3 In this impressive paper, Lu et al have identified specific editing of microRNAs from small RNA sequencing data performed in brain tissues harvested from control and Parkinson's disease (PD) patients. Among the 362 miRNAs identified, they have experimentally validated miR-497-5p which is specifically edited in the prefrontal cortex of PD patients, and identified OPA1 and VAPB as gene targets.
Response 2.3
Thank you.
Comments 2.4. Since I am not an expert, the bioinformatic and statistical analyses should be evaluated by ad hoc referees.
Response 2.4
We are very grateful to you.
Furthermore, as indicated by both reviewers, we carefully improved the language of the manuscript and made many corrections or revisions as listed below.
- On page 1, we corrected the name of the fourth author to “Zhigang Zhao”.
- On page 1, we remove “and” before the tenth author “Jun Yang”.
- On page 1, line 17, “[2] found …” -> “Kim et al. [2] found”.
- On page 1, line 18, “is” -> “was”.
- On page 1, line 20, “are” -> “were”.
- On page 1, line 27, “represses” -> “repressed”.
- On page 1, line 28, “contributes” -> “contributed”; “is” -> “was”.
- On page 1, line 31, “promotes” -> “promoted”.
- On page 1, line 33, “inhibits” -> “inhibited”.
- On page 2, line 47, “4” -> “5”.
- On page 2, line 58-67, we revised the paragraph of the subsection “The small RNA sequencing profiles used”.
- On page 2, line 70, “As in Supplementary Table S1.2” -> “As listed in Supplementary Table S1.2”.
- On page 2, line 76-79, we added a paragraph to introduce the proteomics profiles used in this study.
A previous paper study reported proteomics profiles of prefrontal cortex (BA9) samples in 12 PD patients and 12 healthy people [34]. After comparing the proteomics profiles of the 12 PD patients to those of the 12 healthy people, 283 deregulated proteins were detected (limma package, corrected P < 0.05) and used in our analysis.
- On page 2, line 81, “were” -> “was”.
- On page 3, line 88, “have” -> “had”.
- On page 3, line 104-106, we added a sentence. “and (v) only M/E sites that had significant editing levels in 10% of all the samples were kept in further analysis.”
- On page 3, line 107-110, we added a sentence as below.
Based on the positions of M/E sites in miRNAs and mutations in dbSNP, the identified M/E sites were classified into nine different editing types, i.e., A-to-I, C-to-U, 3’-A, 3’-U, 3’-Other, 5’-editing, Other, SNP and Pseudo [37].
- On page 3, line 117-118, “positions of the sites” -> “positions of the M/E sites in pre-miRNAs”.
- On page 3, line 118-119, we added “in upper cases” and “in lower cases”.
- On page 3, line 119-121, we added a sentence as below.
And edited miRNA was named by the pre-miRNA name, the position of M/E site in pre-miRNA, and the edited/mutated nucleotide in lower case.
- On page 4, line 136, “Supplementary Table S1” -> “Supplementary Table S2.1”.
- On page 4, line 138-139, we added a sentence as below.
The obtained P-values were corrected with the Benjamini and Horchberg method [40] using the mafdr function in MatLab (Mathworks, MA).
- On page 4, line 140, we added “corrected”.
- On page 4, line 143, “U-test” -> “U-tests”.
- On page 4, line 145, “P-value” -> “P-values”.
- On page 4, line 149, “PD-PC and PC samples” -> “PD-PC samples when compared to PC samples”.
- On page 4, line 150, “hcluster” -> “hclust”.
- On page 4, line 153, we added “then”.
- On page 4, line 177, “GO” -> “GO (Gene Ontology)”; “KEGG” -> “KEGG (Kyoto Encyclopedia of Genes and Genomes)”.
- On page 4, line 178, “targets” -> “genes”.
- On page 4, line 180-181, we added “into three major categories, i.e.,”.
- On page 5, line 192-193, “BA9 which was same” -> “BA9 regions which were the same”.
- On page 5, line 211, “with” -> “of”.
- On page 5, line 237, we added “was”.
- On page 5, line 240, “well-received” -> “well received”.
- On page 5, line 241, we added “was”.
- On page 5, line 254, “expression level” -> “expression levels”.
- On page 6, line 260, we added the title of the subsection, “An overview of identified editing sites in miRNAs”.
- On page 6, line 269-270, “Other types” -> “The Other type”.
- On page 6, line 308, we added “identified”.
- On page 8, line 374, “targets” -> “targeted”.
- On page 9, line 434, “this cell line was” -> “the cell lines were”.
- On page 9, line 435, “activity” -> “activities”.
- On page 10, line 443-444, “(Figure 5k-5l)” -> “Figure 5k and 5l, respectively”.
- On page 10, line 455-458, we added the following sentence.
Similarly, we found that the editing levels of three A-to-I editing sites in miRNAs were significantly positively correlated with ages in normal PC samples and these significant correlations were disturbed in PD-PC samples (Figure1d).
- On page 10, line 459, “are” -> “were”.
- On page 10, line 463, “have” -> “had”.
- On page 10, line 465, “is” -> “was”.
- On page 10, line 470, “leads” -> “led”.
- On page 10, line 471, “are” -> “were”.
- On page 10, line 472, “interacts” -> “interacted”.
- On page 10, line 477, “reduce” -> “reduced”.
- On page 10, line 482, “slows” -> “slowed”.
- On page 10, line 483, “promotes” -> “promoted”.
- On page 10, line 485-487, to be more precise, we revised the sentence and added the reference for it as below.
Overexpression wild-type and familial Parkinson’s disease mutant α-synuclein disrupt the interaction between VAPB and PTPIP51 to loosen ER–mitochondria associations [74].
- On page 11, line 491, “a significant inhibitory effect on the proliferation” -> “significant inhibitory effects on the proliferation”.
- On page 11, line 497, “represses” -> “repressed”; “leaded” -> “led”.
- We corrected a mistake when comparing PC and PD-PC samples for identifying M/E sites with significantly different levels in PD-PC samples and the related results. Therefore, we revised Figures 2b-2f, Figure S5a, S5b, Tables S3, S8, and their statements in the main text (mainly on Page 7 to 8, lines 324 – 357).
- We also revised the legends of Figures 1, 2, 3 and 5.
- We revised Supplementary Figures S2 to S6.
- We enlarged the figures to make them more legible.

Reviewer 2 Report
this manuscript contains a huge amount a data analysis and develops noval approaches to investigate the impact of miRNA editing on gene expression in the fram of Parkinson's disease.
The only limitation that I see would be, in the introduction, to provide only the relevant examples of miRNAs likely involved in PD, rather that this exhaustive list with 13 references. Otherwise, the paper is very impressive.
In this impressive paper, Lu et al have identified specific editing of microRNAs from small RNA sequencing data performed in brain tissues harvested from control and Parkinson's disease (PD) patients. Among the 362 miRNAs identified, they have experimentally validated miR-497-5p which is specifically edited in the prefrontal cortex of PD patients, and identified OPA1 and VAPB as gene targets.
Since I am not an expert, the bioinformatic and statistical analyses should be evaluated by ad hoc referees
Author Response

(The authors gave the same response as above.)

Round 2
Reviewer 1 Report
I thank the authors for the response. Indeed, since the title “Characterizing relevant microRNA editing sites in Parkinson’s disease “ is about PD, the mRNAs in glioma cell lines that would be regulated by p497 and p497_25g are not necessary the relevant ones . However, from a mechanistic point of view this would prove the shift of targets by the editing and provide much more reliable experimental evidence than the reporters. I would not object if the editor wants to accept the paper as is though. It is an overall very well written article and provides sufficient analysis to support the conclusions made.